# Rain Cover and Netting Materials Differentially Affect Fruit Yield and Quality Traits in Two Highbush Blueberry Cultivars via Changes in Sunlight and Temperature Conditions

**DOI:** 10.3390/plants12203556

**Published:** 2023-10-13

**Authors:** María F. Matamala, Richard M. Bastías, Ignacio Urra, Arturo Calderón-Orellana, Jorge Campos, Karin Albornoz

**Affiliations:** 1Departamento de Producción Vegetal, Facultad de Agronomía, Universidad de Concepción, Chillán 3780000, Chile; mariamatamala@udec.cl (M.F.M.); iurra@udec.cl (I.U.); arcalderon@udec.cl (A.C.-O.); 2Departamento de Producción Animal, Facultad de Agronomía, Universidad de Concepción, Chillán 3780000, Chile; jcamposp@udec.cl; 3Department of Food, Nutrition, and Packaging Sciences, Coastal Research and Education Center, Clemson University, 2700 Savannah Highway, Charleston, SC 29414, USA; kpalbor@clemson.edu

**Keywords:** protected fruit growing, UV light, thermal accumulation, plant growth, fruit firmness, *Vaccinium corymbosum* L.

## Abstract

The use of covers to protect blueberry orchards from adverse weather events has increased due to the variability in climate patterns, but the effects of rain covers and netting materials on yield and fruit quality have not been studied yet. This research evaluated the simultaneous effect of an LDPE plastic cover, a woven cover, and netting material on environmental components (UV light, PAR, NIR, and growing degree days (GDDs)), plant performance (light interception, leaf area index, LAI, yield, and flower development), and fruit quality traits (firmness, total soluble solids, and acidity) in two blueberry cultivars. On average, UV transmission under the netting was 11% and 43% higher compared to that under woven and LDPE plastic covers, while NIR transmission was 8–13% higher with both types of rain covers, with an increase in fruit air temperature and GDDs. Yield was 27% higher under the woven cover with respect to netting, but fruit firmness values under the netting were 12% higher than those of the LDPE plastic cover. Light interception, LAI, and flower development explained 64% (*p* = 0.0052) of the yield variation due to the cover material’s effect. The obtained results suggest that the type of cover differentially affects yield and fruit quality in blueberries due to the specific light and temperature conditions generated under these materials.

## 1. Introduction

A wide variety of blueberry (*Vaccinium corymbosum* L.) cultivars are suitable for cultivation in a vast area of Chile, and they are mainly cultivated in the Maule, Ñuble, and Araucanía regions [1]. One of the most planted cultivars is ’Legacy’ [1], which belongs to the group of southern highbush blueberries, characterized by their low chill requirement (500–600 chill hours) during winter dormancy [2]. In recent years, ‘Legacy’ has been replaced by other cultivars that better meet the industry requirements in terms of fruit quality, flavor, and firmness, and that belong to the most demanding group of northern highbush blueberries, with requirements of 800–1000 chill hours [3].

Given the current scenario of climate change and the need to expand market opportunities for exported fruit, the production of blueberries under protected cultivation has become widespread worldwide. The most commonly used protection systems are roof covers and high tunnels, which protect crops from rain and frost, in turn accelerating fruit maturity and advancing the harvest date [4]. Furthermore, netting is also an effective tool to protect orchards from sunburn, hailstorms, and insect attacks [5].

The most commonly used materials in rain-protection systems are waterproof woven covers with a laminated texture or low-density polyethylene (LDPE) plastic covers with a smooth texture, while porous and permeable raschel or monofilament nets are used for netting [5,6]. Ogden and van Iersel [7] evaluated LDPE plastic covers for ‘Emerald’ and ‘Jewel’ blueberry cultivars and concluded that this type of cover affected the synchronization of flowering and pollination, thus decreasing fruit set and yield. Conversely, other studies on the effect of LDPE plastic covers on blueberry cultivars have reported that yields of ‘O’Neal’ and ‘Legacy’ increased by over 40% [8], while no effects were observed in ‘Sampson’ and ‘Duke’ [9]. In ‘Bluegold’ and ‘Brigitta’ blueberries, the use of woven covers decreased yields by 28% and 73% compared to non-covered plants [10]. Regarding netting, Retamales et al. [11] found that the use of white and red nets increased the yield of ‘Berkeley’ blueberries by 84.2 and 31.9%, respectively, reporting no effects on fruit size or the content of soluble solids in the fruit. Likewise, Lobos et al. [12] evaluated the effect of black, red, and white nets with different shade intensities on ‘Elliott’ blueberries, concluding that red and white nets with intermediate shade intensities delayed harvest without affecting yield or fruit quality.

Therefore, there is evidence that protection covers have an impact on the yield and quality of blueberries, with varying effects depending on the cover material and cultivar. However, there is little information about the environmental factors that determine differences between types of covers, and there are few studies that have analyzed different cover materials simultaneously for blueberries. It has been demonstrated that the specific characteristics of the cover material, in terms of color and pattern, determine variations in the quantity and composition of the light radiation transmitted by these materials [13,14], as well as variations in the coefficients of heat transfer, which directly impact the environmental temperature [15]. Depending on the color and thread density, cover materials alter light transmission in the UV (280–390 nm) and PAR (400–700 nm) ranges. Thus, the use of translucent nets reduces the transmitted PAR by up to 7%, while black nets result in an 18% reduction. In addition, netting can reduce UV light transmission by 10–13% more than PAR transmission [5]. On the other hand, Salazar-Canales et al. [16] determined that blue-gray, black, and pearl-grey nets reduce radiation by 24%, 21%, and 19%, respectively.

Regarding waterproof materials, LDPE plastic reduces PAR transmission by 15% and transmits 4% UV radiation. Likewise, it has been described that this material transmits 7% more PAR radiation on sunny days than woven covers, with no differences between the materials on cloudy days [17]. On the other hand, Abdel-Ghany et al. [15] found differences in heat transfer between different colored nets, reporting that green nets increased the convection heat transfer coefficient by 37.8%, while beige nets reduced this coefficient by 35.4% compared to dark green and white nets. Increases in maximum air temperature have also been recorded in polyethylene high tunnel-covered blueberry orchards, with increases between 3 °C and 15 °C when compared to non-covered plants [7]. The recent advances in machine learning algorithms have proven to be useful in predicting yield and quality in blueberry crops that grow in dynamic environments based on mathematical models [18], which could be perfect for use in blueberry crops protected under cover. Nevertheless, to incorporate this tool in protected blueberry orchards, more knowledge is necessary about how environmental conditions under different types of cover materials affect yield and fruit quality aspects in blueberry plants. The present study proposes that the materials used in rain protection and netting cover materials differentially influence the yield and fruit quality of blueberries by modifying the light and temperature conditions generated by these crop protection systems. To test this hypothesis, the objective of this work was to evaluate the effects of LDPE plastic covers, woven covers, and netting on the quantity and quality of solar radiation, as well as temperature variations and accumulation, to determine their impact on plant performance (flower development, fruit set, yield, leaf area index) and fruit quality traits (size, firmness, total soluble solids, and acidity) in southern highbush (‘Legacy’) and northern highbush (‘Top Shelf’) blueberries with low and high chill requirements, respectively.

## 2. Results

### 2.1. Woven and LDPE Covers Boost Yield by Enhancing Microclimate and LAI and Reducing UV Exposure

PAR transmission (%) showed no significant differences (*p* = 0.154; Figure 1) between the different cover materials. However, significant differences (*p* < 0.0001) were found in terms of UV transmission. Netting transmitted 76.7% of the external UV radiation, while LDPE plastic and woven covers transmitted lower UV levels, with reductions of 70.6% and 19.5% with respect to netting, respectively.

Radiation flux partitioning obtained by spectrophotometric analysis showed that the UV radiation transmitted on average during the season in the two locations, Linares and Traiguén, was 53%, 42%, and 10% under netting, woven covers, and LDPE plastic covers, respectively (Figure 2A,B); PAR transmission reached 83%, 81%, and 86% (Figure 2C,D), while NIR transmission reached 83%, 91%, and 96%, under the same cover systems, respectively (Figure 2E,F).

All the covers decreased the photosynthetic photon flux density (PPFD) transmitted to the plant for ‘Legacy’ and ‘Top Shelf’ in both locations, with greater magnitude and significance from 80 cm to 140 cm from the center of the plant to the middle of the inter-row (Figure 3 and Figure 4). In Linares, the proportion of transmitted PPFD was reduced by 46.7%, 37.9%, and 22.6% in ‘Legacy’ under woven covers, LDPE plastic covers, and netting, respectively (Figure 3A). In ‘Top Shelf’, the transmitted PPFD decreased by an average of 43.6%, 32.1%, and 21.1% under LDPE plastic covers, woven covers, and netting, respectively (Figure 3B). In Traiguén, PPFD transmission decreased by an average of 36.8%, 31.7%, and 22% in ‘Legacy’ (Figure 4A), while for ‘Top Shelf’, reductions of 36.4%, 32.2%, and 20.5% were recorded under the same protection covers, respectively (Figure 4B).

In terms of the leaf area index (LAI) (Table 1), values were significantly higher (*p* < 0.0001) under woven and LDPE plastic covers with respect to the control (no cover), with increases of 39.3% and 38.8% in Linares and 50.5% and 54.9% in Traiguén, respectively (Table 1). When compared to the control, light interception was also significantly (*p* < 0.0001) higher under woven and LDPE plastic covers, reaching increases of 44.3% and 43.6% in Linares and 52.6% and 58.2% in Traiguén, respectively. In Linares, the average values of light interception of blueberry plants grown under woven and LDPE plastic covers were 10.2% and 20.3% higher than that of netting and the control, respectively, while increases of 8.1% and 20.6% were observed in Traiguén (Table 1). In Linares, the average LAI values observed under LDPE plastic and woven covers were 20.7% and 39.1% higher than those of netting and the control, while Traiguén recorded increases of 21. 1% and 52.7%, respectively. In addition, light interception and LAI values were significantly higher in ‘Legacy’ compared to ‘Top Shelf’ in Linares, with increases of 8% and 17.3%, respectively (Table 1).

Cover material had a significant impact (*p* < 0.0001) on yield in Linares and Traiguén (Table 2 and Table 3). In both locations, blueberry plants grown under woven covers had higher fruit yields compared to plants grown under netting and no cover (control), with increases of 24.2% and 18.1% in Linares and 31% and 13.7% in Traiguén, respectively. The woven cover also resulted in higher yields than the LDPE plastic cover, but only in Traiguén. In addition, netting led to significantly lower yield in both locations, with reductions of 18.8% and 20.8% in Linares and Traiguén, respectively (Table 2 and Table 3). Regarding cultivars, fruit yield was significantly higher (*p* = 0.0004 and *p* = 0.0155 in Linares and Traiguén) for ‘Top Shelf’ compared to ‘Legacy’, at 23% and 10.4% greater in Linares and Traiguén, respectively (Table 2 and Table 3).

No significant differences were observed in terms of flower development due to the effect of covers in both locations (Table 4). However, the number of floral primordia per bud was significantly affected by the effect of the cultivar; this was 25% (*p* < 0.0001) and 17% (*p* = 0.0003) higher in ‘Top Shelf’ than ‘Legacy’ for Linares and Traiguén, respectively. In Linares, fruit set was not significantly affected by the cover material or cultivar. In Traiguén, the plants under netting exhibited a significant increase (*p* = 0.0183) of 11% in fruit set with respect to the control (Table 4).

### 2.2. UV-Exposed Control and Netting-Covered Fruits Showed Increased Firmness; LDPE and Woven Covers Decreased Firmness

Fruit size, measured as the diameter of the fruit, was not significantly affected by cover material (Table 2 and Table 3). However, the cultivar had a significant impact on this parameter, with ‘Top Shelf’ exhibiting the best performance and reaching values that were 11.4% and 21.6% higher than those recorded by ‘Legacy’ in Linares and Traiguén, respectively (Table 2 and Table 3).

Fruit firmness (in g mm^−1^) was affected by the cover material (Table 2 and Table 3). Netting presented significantly (*p* < 0.0001) higher values than the control (no cover) in Linares and Traiguén (3.6% and 4% higher, respectively). Conversely, the woven cover resulted in significant decreases (*p* < 0.0001) of 3% and 6.3% with respect to the control for Linares and Traiguén, respectively. Furthermore, significantly (*p* < 0.0001) lower values were observed in plants grown under LDPE plastic covers, with values that were 6% and 9.2% lower than those recorded for the control for Linares and Traiguén, respectively (Table 2 and Table 3).

### 2.3. Netting Maintains Firmness by Lowering Air Temperature

Fruit air temperature differences under LDPE plastic covers were significantly higher than those recorded under netting or a woven cover, reaching a value of +0.3 °C, while fruit air temperature differences under these materials were significantly reduced by −0.3 °C and −0.1 °C, respectively (Figure 1C). Accumulated growing degree days (GDDs) during the season were 46% higher in Linares compared to Traiguén (Figure 5). In Linares, the LDPE plastic cover increased the amount of GDDs by 17% compared to the control (no cover), followed by the woven cover and netting with 10% and 8%, respectively (Figure 5A). In Traiguén, the use of netting and a plastic cover reduced the accumulation of GDDs by about 11%, while the woven cover increased the amount of accumulated GDDs by 3% compared to the control (Figure 5B).

### 2.4. Soluble Solids Exhibit Variations Based on Cover Type and Cultivar, Impacting Fruit Composition

The concentration of soluble solids of the fruits (measured as °Brix) was also affected by cover materials (Table 2 and Table 3). In Linares, the fruits grown under an LDPE plastic cover presented a significantly higher value (*p* < 0.0001) than the control (1.4% higher), while the fruits grown under netting recorded values that were 2.5% lower than the control (Table 2). In Traiguén, all the cover materials significantly reduced (*p* = 0.0066) the content of soluble solids in the fruit, at 3.7%, 2.5%, and 2.3% lower in blueberries grown under LDPE plastic covers, woven covers, and netting, respectively (Table 3). It is interesting to note that no significant differences were observed in terms of acidity content or the total soluble solids to acidity ratio, either due to the effect of cover material or cultivar (Table 2 and Table 3).

### 2.5. Achieving Optimal Quality and Yield through Empirical Cover, Cultivar, and Material Selection

In Traiguén, there was a significant effect (*p* < 0.0001) of the cultivar on fruit firmness, at 13.2% higher in ‘Top Shelf’ compared to ‘Legacy’ (Table 3). In the same location, a significant effect (*p* = 0.0007) of the interaction of the cover material with the cultivar was also observed (Figure 6). The variety ‘Top Shelf’ grown under netting had the highest value of fruit firmness, at 5.8% higher than the control. Conversely, significantly lower values were observed in the interaction of ‘Legacy’ with the LDPE plastic cover, with firmness being 12.8% and 20% lower than that observed in non-covered plants of ‘Legacy’ and ‘Top Shelf’, respectively (Figure 6). In addition, there was a significant effect (*p* = 0.0179) of the interaction of the cover material with the cultivar on soluble solids in Traiguén (Figure 7). The interactions of ‘Top Shelf’ with LDPE plastic cover and ‘Legacy’ with netting presented significantly lower values of soluble solids in the fruit, decreasing by 4.3% and 4.6%, respectively, with respect to the control (Figure 7). In addition, the concentration of soluble solids was significantly affected (*p* = 0.0055) by the cultivar only in Linares, at 2.7% higher in ‘Top Shelf’ compared to ‘Legacy’ (Table 2).

A significant effect of the interaction of the cover material with the cultivar on yield was observed in both locations (*p* = 0.0011 and *p* = 0.0500 in Linares and Traiguén, respectively). In Linares (Figure 8), the interaction of ‘Top Shelf’ with woven and LDPE plastic covers led to significantly higher yields in relation to the other combinations, at 25.2% and 21.4% higher with respect to non-covered ‘Top Shelf’ plants and 35.9% and 31.8% higher than ‘Top Shelf’ plants under netting. In Traiguén (Figure 9), the interaction of ‘Top Shelf’ with the woven cover led to a significantly higher yield than the other combinations, at 13.5% and 42% greater than that of this cultivar without cover and under netting, respectively. In both locations, the interaction of ‘Legacy’ and ‘Top Shelf’ with netting resulted in significant reductions in yield, at 32.7% and 26.4% lower with respect to ‘Top Shelf’ plants under a woven cover in Linares, and 28.2% and 29.6% lower than ‘Top Shelf’ under a woven cover in Traiguén (Figure 8 and Figure 9).

According to the multiple linear regression analysis, flower development and LAI significantly explained (*p* = 0.0052) 63% of the variation in yield due to the effect of ‘Top Shelf’ cultivar and the woven cover (Figure 10a), since the highest yield values corresponded mainly to this interaction in both locations (Figure 8 and Figure 9). In turn, light interception and flower development significantly explained (*p* = 0.0014) 64% of the variation in yield due to the effect of the same interaction (Figure 10b). In quantitative terms, the optimal ranges to achieve high yields consist of flower development greater than 6 primordia per bud and an LAI greater than 1.5 (Figure 10a). Likewise, the highest yields under cover can be obtained in plants with buds with more than 6 flower primordia per bud and with a light interception capacity greater than 50% (Figure 10b).

## 3. Discussion

### 3.1. UV Light Exposure Effects on Yield and Fruit Firmness

Blueberries grown under LDPE plastic and woven covers had a significantly higher yield compared to those under netting in both locations (Table 2 and Table 3). These results could be attributed to the light microclimate under woven and LDPE plastic covers (Figure 1 and Figure 2). In this sense, it has been described that plant growth and leaf development increase due to reduced light levels, which is known as shade avoidance syndrome, as a response to a reduction in the red-to-far-red light ratio mediated by phytochromes, a decrease in the blue-to-red light ratio mediated by cryptochromes, or by a decrease in UV light mediated by the action of a specific UVR8 receptor, which is activated or deactivated depending on the intensity of UV-B light [19]. In our research, there was a significant increase in LAI in both Linares and Traiguén for plants under woven and plastic covers (Table 1), which were the materials that most effectively blocked UV radiation (Figure 1 and Figure 2). These results coincide with previous studies in eggplant and pepper crops, where the use of UV-blocking covers resulted in an increase in stem length and plant height. In plants such as chrysanthemum (*Chrysanthemum indicum* L.), there was also an increase in plant height under UV-blocking covers due to a greater number of internodes [20]. On the other hand, leaf area and dry matter increased in cucumbers (*Cucumis sativus* L.), broccoli seedlings (*Brassica oleracea* L. var. italica), and turnips (*Brassica rapa* L.) grown under protective covers with decreased UV transmission [20]. Similarly, another study showed that high UV radiation reduced the leaf area in blueberry plants by decreasing the number of buds and leaves [21]. The LAI, defined as m^2^ of leaves over m^2^ of land, determines the relationship between light interception and yield; thus, a rapid increase in LAI is desirable in young orchards to allow for greater light interception for photosynthesis and assimilate partitioning, which significantly increase yield [22,23]. In the present study, the PPFD intercepted by blueberry plants was favored by an increase in LAI under woven and LDPE plastic covers (Table 1), enhancing the availability of PAR light for plant photosynthesis, which directly favors the yield potential of the crop [24]. This would explain why blueberry plants under woven and LDPE plastic covers had higher yields, which coincides with previous studies in which specific conditions of the low-red/far-red light ratio under covers favored a greater development of leaf area by phytochrome action, thus allowing a greater capacity to intercept light for photosynthesis in young apple plants (*Malus domestica* Borkh.). A positive impact on dry matter yield and fruit growth under this type of cover material was also noted [25].

Light transmission under woven and LDPE plastic covers was lower compared to values observed with netting and the control (Figure 3 and Figure 4). This demonstrates that, when these types of covers are used, the PAR reaching the soil surface is lower in the sections closest to the inter-row. Based on discontinuous canopy, this indicates that the plants grown under these covers would present greater uptake of PPFD through the canopy, which is directly dependent on the increase in LAI [26]. In addition to light interception, blueberry yield is also determined by the efficiency of converting light into biomass by the plant, which largely depends on the photosynthetic capacity of the leaves [27]. In blueberry plants, it has been determined that a high incidence of UV radiation can cause damage at the cellular level, affecting the integrity of the thylakoid membrane and the photosystem II (PSII) and decreasing the net assimilation of CO_2_ [28], thereby reducing the photochemical efficiency of PSII and net photosynthesis [29]. In other fruit species such as mango (*Mangifera indica* L.), the increase in UV radiation decreases the leaf transpiration rate and stomatal conductance and resistance, reducing the intercellular CO_2_ concentration, affecting CO_2_ assimilation, and resulting in a decrease in photosynthesis caused by stomatal restriction, with a negative impact on yield and fruit quality [30]. Even though the present study did not evaluate photosynthetic aspects of the leaf, the fact that leaf development was affected by differences in UV radiation transmitted by cover materials indicates that leaf photosynthetic aspects may also be affected, which requires further investigation.

The differences in fruit firmness (Table 2 and Table 3) observed with the use of different covers could also be associated with UV light exposure. Martin and Rose [31] described that the cuticle provides protection against excessive sunlight, and that fruits exposed to higher UV radiation, which is particularly harmful, have a thicker cuticle as a defense mechanism. During the development and ripening of tomatoes, protection against UV radiation is also enhanced by cuticle thickening and the accumulation of phenolic compounds [32]. In grapes, the accumulation of cuticular waxes is significantly higher in fruits exposed to full sun compared to shaded fruits [33]. Furthermore, increased cuticle thickness has also been observed in blueberry fruits exposed to the sun [34].

### 3.2. The Significance of Cuticle Thickness in Regulating Fruit Firmness

Apart from being a physical barrier that protects plants and fruits from biotic and abiotic stresses, the cuticle also has a mechanical function and provides protection against fruit bruising [35]. In fact, this membrane provides structural support for fruits lacking hard internal tissue, as it is an external structural element that adds mechanical support for tissue integrity and thus plays an important role in fruit firmness during harvest and postharvest [36]. In the present study, fruits were significantly firmer under higher UV radiation levels, as observed in the control (no cover) and with netting, while fruits grown under covers with a lower UV light transmission capacity, such as the LDPE plastic cover, presented lower firmness (Table 2 and Table 3; Figure 1 and Figure 2). These differences could be attributed to changes in fruit cuticle thickness and should be studied in future research.

### 3.3. Temperature’s Role in Influencing Yield and Fruit Firmness

Temperature is another factor that can influence the yield of blueberries grown under covers. Different studies have shown that GDDs are linearly correlated with shoot growth and leaf area per shoot in species such as apple [37], cucumber (*Cucumis sativus* L.), and sweet pepper (*Capsicum annuum* L.), and are a good predictor of LAI in crops [38]. GDD accumulation reached higher values with woven and LDPE plastic covers in Linares (Figure 5A), where the highest LAI and light interception values were recorded (Table 1). In Traiguén (the location with the lowest GDD accumulation), however, the studied covers showed no clear effect on this measure, except for the woven cover. It has been described that the effect of covers on temperature can vary depending on local environmental conditions. Accordingly, differences in heat loss due to local weather conditions impact the temperature of buds and leaves [39]. In the present study, both LDPE plastic and woven covers increased the fruit temperature to above air temperature. However, this behavior was more stable in terms of GDDs for warmer conditions like those of Linares (Figure 1 and Figure 5).

Temperature variations could also explain the differences in fruit firmness due to the effect of cover materials. NIR transmission under LDPE plastic and woven covers was higher than that under netting (Figure 2E,F), which was also reflected in the difference between fruit and air temperatures with these materials (Figure 1C). It has been determined that an increase in temperature above 30 °C negatively affects fruit firmness in blueberries [40]. This has also been reported in species such as cherry [41], grape [42], avocado [43], and apple [44]. Being a climacteric fruit, changes in fruit firmness in blueberries are mainly related to water loss due to respiration and transpiration processes, mainly triggered by a temperature increase [45]. Fruit softening is also associated with cell wall hydrolysis, activated by enzymes that depolymerize components, and whose transcription can be induced by heat stress [46]. Therefore, it seems that the temperature under the cover also plays a role in fruit firmness. However, this was a partial effect, only observed in Linares, where the use of LDPE plastic and woven covers (decreased fruit firmness) increased the accumulation of GDDs with respect to the control and netting (Figure 5). In both Linares and Trainguén, however, netting always presented the lowest GDD values with respect to the control or the other cover materials, which is explained by a greater capacity to block NIR light and reduce fruit temperatures (Figure 1 and Figure 2), probably because of the benefits of black shade netting for plants. In fact, black nets have a greater capacity to decrease the air temperature compared to other colors [15], which would also explain why fruits were significantly firmer under this type of cover compared to the others (Table 2 and Table 3).

### 3.4. Genetic Variability and Cultivar-Specific Responses

On the other hand, the interaction of woven cover with ‘Top Shelf’ resulted in the highest yield in both locations (Figure 8 and Figure 9), indicating that the yield of blueberries depends on internal factors such as genetics and external factors such as management practices and the climate, as previously reported in this crop [47]. Given that the fruit diameter and number of flower primordia per bud were significantly higher for ‘Top Shelf’ (Table 2 and Table 3), and also considering that plants under woven covers presented higher GDD values compared to non-covered plants in both Linares and Trainguén, the results of the present study indicate that the higher yield achieved by ‘Top Shelf’ under a woven cover can be explained by the interaction between genetic and environmental components. The former corresponds to fruit size and the fertility of flower buds, and the latter corresponds to lower transmission of UV radiation and greater accumulation of GDDs, which favor a greater LAI and PPFD interception under these particular light and temperature conditions. In fact, this was confirmed through a multiple linear regression analysis (Figure 10), demonstrating that the highest yield values are obtained in specific ranges of number of flower primordia per bud, LAI, and light interception, the variables of which could explain more than 60% of the variation in crop yield of both blueberry cultivars under the three types of covers evaluated in this research. This type of analysis has also been applied to other fruit species such as cranberry (*Vaccinium macrocarpon* Ait.), demonstrating that variables such as light and temperature allow a prediction of fruit growth and yield [48]. Similarly, there is evidence that the number of flower buds in blueberries is a good predictor of the number of fruits, while variables related to light interception, LAI, and flower primordia per bud are also strongly correlated with yield [27,47]. Therefore, according to previous research and our results, this suggests that it is possible to develop predictive models of yield for different blueberry cultivars grown under different types of covers based on the quantification of variables related to flower development, LAI, and light interception of plants grown under these environmental conditions.

It is important to note that the interaction of ‘Top Shelf’ with netting led to the highest fruit firmness values in Traiguén (Figure 6), while ‘Top Shelf’ fruits presented higher fruit firmness compared to ‘Legacy’ in both locations (Table 2 and Table 3). This would indicate that the genetics of the crop influence the response to higher levels of UV light or lower temperatures under netting as an adaptation mechanism to heat stress, increasing cuticle thickness as an external structural support. The latter is also due to improved temperature conditions that allow reinforcement of the cell walls and internal structural support; therefore, differences in the chemistry of the membranes could give rise to differences in heat and UV radiation tolerance between cultivars [34]. In blueberries, the composition of the cuticle varies depending on the cultivar, allowing for a certain heat or solar radiation tolerance thanks to the different composition of membrane lipids [49]. Accordingly, it would be interesting to study these physiological and biochemical aspects of fruits and evaluate crops under cover materials with different light transmission capacities in the UV and NIR spectra, as this could help select or develop materials to achieve the highest fruit firmness according to the cultivar or climatic conditions. This is particularly important considering that firmness is an attribute that determines the quality of fruits, including blueberries [36].

### 3.5. Implications for Selecting Suitable Cover Materials

The amount of sunlight under which blueberry leaves achieve their highest photosynthesis for adequate growth and development is a PPFD between 700 and 900 µmol m^−2^ s^−1^. With PPFD values greater than 1000 µmol m^−2^ s^−1^, blueberry leaves lose their capacity to increase photosynthesis, even decreasing it due to possible inhibition of this process due to excess light. On the other hand, and depending on the cultivar, the optimal temperature range for blueberry growth ranges between 20 and 26 °C, while in temperature conditions above 30 °C, the photosynthesis capacity of the plant is reduced, diminishing the total plant biomass [27]. Since cover materials alter PPFD availability and thermal accumulation, functional–structural models could be developed to predict the growth and development of blueberries in protected environments. These models are based on simple morphological traits such as leaf number and specific leaf length and how they are affected by temperature and photosynthetic light availability under greenhouses [38]. They could be perfect for the prediction of yield in blueberries under different types of cover materials.

### 3.6. Exploring Variability in Total Soluble Solids and Its Environmental Implications

Finally, the effect of cover materials on total soluble solids in fruits varied between locations. Although cover materials have a significant effect on this quality trait, the type of cultivar also played a role, resulting in greater variability of the results (Figure 7). Synthesis, degradation, and translocation of sugars and organic acids at ripening stage cause changes, resulting in differences depending on the genetic origin of these processes [50]. Previous studies conducted on apple trees reported a great variability in the concentration of soluble solids between cultivars under covers, concluding that this quality trait is often more influenced by the environmental conditions in each growing season, promoting typical responses to shade under netting in the presence of variations in light and temperature conditions [51], which could explain the results of this research on blueberries.

## 4. Materials and Methods

### 4.1. Study Sites

The present study was carried out during the 2021–2022 season. The experiment was repeated in two locations in central-southern Chile with different environmental conditions: Linares, Maule region (35°49′4.34″ S 71°32′26.91″ W), and Traiguén, La Araucanía region (38°19′52.62″ S 72°41′35.47″ W). Linares is located in the Central Valley, characterized by a warm temperate climate with a dry subhumid moisture regime. The average annual rainfall is 1137 mm, with a dry period of 5 months. The maximum temperature reaches 29.1 °C in January and the minimum temperature goes down to 3.5 °C in July [52]. Traiguén has a warm temperate mesothermal climate, with a dry subhumid moisture regime. The average annual precipitation is 1133 mm, with a dry period of 5 months. The maximum temperature reaches 26 °C in January and the minimum temperature is 4.1 °C in July [52]. The soil texture is clay loam and silty clay in Linares and Traiguén, respectively.

### 4.2. Plant Material and Experimental Design

This study was conducted on ‘Legacy’ and ‘Top Shelf’ blueberry cultivars. In Linares, the plants were established in October 2018, with a plant spacing of 3 m between rows and 1 m between plants. In Traiguén, the plants were established in November 2017, with a plant spacing of 3.2 m between rows and 1 m between plants.

Both cultivars were protected by three different covers: a high-density laminated woven cover (Agrosystems S.A., Santiago, Chile); an LDPE plastic (Agrosystems S.A., Chile) cover; and black monofilament net at 20% shade (Delsantek S.A., Santiago, Chile) (Figure 11). A control treatment (no cover) was also included.

The covers were installed on a roof-type structure with a height of 3 m above the ground, a width of 2.5 m, and an inclination angle from the roof edge of 28°. These were used from the beginning of flowering (August 15) to the beginning of leaf fall (April 15), covering three rows per plot of 12 and 15 plants in Linares and Traiguén, respectively (Figure 12). In both locations, the experiment was conducted in a completely randomized block experimental design with a divided plot arrangement, with four replicates and two plants as the experimental unit. The cover materials were the main plot and cultivars were the subplot.

### 4.3. Light and Temperature Conditions

An SP-110 pyranometer (Apogee, Logan, UT, USA) was installed at each location to continuously record the variation in global solar radiation (W m^−2^) in the range of 360–-1120 nm; the information was stored in an Em50 datalogger (Decagon Devices, Pullman, WA, USA), and was recorded during the whole period for which covers were installed. The radiometric characteristics of the cover materials were evaluated according to the methodology proposed by Olivares-Soto et al. [53]. For this, a 1 × 1 m sample of material was placed at a height of 1.5 m from the ground and the spectral light transmission was determined in full sun at solar noon (12:30–13:30), and in the same wavelength range of the pyranometer. For the measurement, UV-VIS-IR spectrophotometers (models BLUE-Wave and DWARF-Star) were connected to a CR2 cosine receptor (StellarNet INC., Tampa, FL, USA). Simultaneously, transmission of photosynthetic light (PAR, µmol m^−2^ s^−1^) and ultraviolet light (UV, W m^−2^) was estimated using an MQ-200 quantum sensor (Apogee Instruments Inc., Logan, UT, USA) and an ultraviolet MU-250 sensor (Apogee Instruments Inc., Logan, UT, USA), respectively; measurements were randomly repeated three times. Based the information obtained, solar radiation flux partitioning in the range of UV light (360–399 nm), PAR light (400–700 nm), and NIR light (701–1120 nm) was estimated under the different cover materials in field conditions and using the coefficients of transformation of energy to radiant flux proposed by Nobel [54].

Once harvest was finished, the photosynthetic photon flux density (PPFD, µmol m^−2^ s^−1^) was measured using an AccuPAR LP-80 ceptometer (Decagon Devices, Pullman, WA, USA) according to the methodology proposed by Wünsche et al. [55]. For this, two hours before solar noon (11:00), at solar noon (13:00), and two hours after solar noon (15:00), the ceptometer rod was passed at ground level and under the canopy of the plant and from the midpoint of the inter-row to the other midpoint of the following row and every 20 cm, leaving the unitary sensor of the ceptometer in full sun as a reference. Based on the information obtained, the amounts of PPFD transmitted (%) by the plant, light interception (%), and leaf area index (LAI) were determined under the different covers.

Simultaneously, via spectrophotometry measurements, the temperature emission capacity of the cover materials was estimated, following the methodology proposed by Abdel-Ghany et al. [15]. For this, the skin temperatures of an apple fruit (°C) and the air (°C) were measured after being exposed to full sun at a distance of 30 cm from the cover material. Temperature measurements were made with an SI-111-SS infrared radiometer (Apogee Instruments Inc., Logan, UT, USA), which was placed 10 cm from the fruit and pointing directly at the fruit surface. The information obtained was used to estimate the fruit–air temperature difference (°C) under each cover; measurements were repeated three times. At the field level, the air temperature (°C) was recorded at 15 min intervals using iButton DS1923 meteorological sensors (Maxim/Dallas Semiconductor Inc., Dallas, TX, USA), which were installed inside a screen sun protection at a height of 1.5 m above ground level. The accumulation of GDDs (base 10 °C) was calculated according to the methodology proposed by McMaster and Wilhelm [56].

### 4.4. Yield Components and Fruit Quality Traits

During the winter recess and before pruning, a sample of three shoots per plant was taken and flower development was evaluated (number of flower primordia bud^−1^) using a stereomicroscope (Olympus^®^, model SZ61, Tokyo, Japan) equipped with a digital camera (Olympus^®^, LC30 Tokyo, Japan). The fruit set was estimated during the flowering stage. For this, three shoots per plant were marked and flowers per cluster were counted; after 3 weeks, fruits were counted and thus the fruit set percentage (%) was determined.

At harvest, all the fruits were picked and the accumulated yield in kg plant^−1^ was determined using a precision balance model PCE-PCS 30 (PCE Instruments, Santiago, Chile). In total, 6 harvesting events were carried out from 3 December 2021 to 4 January 2022 in Linares and from 23 December 2021 to 27 January 2022 in Traiguén.

For each of the harvest events, a sample of 20 fruits was taken and the fruit diameter (mm) and firmness (g mm^−1^) were measured using a Firmpro texturometer (HappyVolt, Santiago, Chile). For this, each fruit was placed with the cheek side on the texturometer tray and was compressed once by a flat probe at a pressure force of 800 g (with an error of less than 0.35 g) and at a spatial resolution of 0.0025 mm (with a spatial error of less than 0.04 mm). Subsequently, the content of soluble solids (SS, °Brix) and the acidity (A, % citric acid) were determined using a digital refractometer model PAL-BX ACD1 Master Kit (ATAGO, Tokyo, Japan). For this, the initial sample of 20 fruits was taken and the total soluble solids was measured in each berry. Next, the 20 fruits were crushed to obtain 10 g of juice, to which 40 g of distilled water was added, forming a solution as a composite sample to measure acidity. Then, the SS/A ratio was estimated.

### 4.5. Data Analysis

Data were analyzed by an analysis of variance (ANDEVA), while the normality of residuals and the homoscedasticity of the variance were previously tested. Differences between means were determined using a Tukey’s test with a significance level of 0.05. In order to find a linear relationship between the dependent variable, Y = yield, and the independent variables (X) (light and plant performance), a multiple linear regression model was applied, including dependent variables as fixed effects and locations as random effects, with a coefficient of determination (R^2^) at a significance level of 0.05. All the analyses were performed with Infostat [57] and R [58] software using the “agricolae” package, version 1.3–5 [59].

## 5. Conclusions

The results obtained suggest that the type of cover material accounts for differences in the yield of ‘Top Shelf’ and ‘Legacy’ blueberry cultivars in Linares and Traiguén. This is directly related to the decreased UV light transmission and increased accumulation of GDDs observed with the use of woven and LDPE plastic covers, causing an increase in LAI as a plant response to such conditions, as well as greater light interception, with a positive impact on yield. On the other hand, differences between cover materials in terms of UV and NIR transmission and accumulation of GDDs also had an impact on fruit firmness. A greater firmness was observed under netting, probably due to the effect of increased UV radiation; conversely, significantly lower values were recorded in fruits grown under the LDPE plastic cover due to the reduction in UV light transmission or the significant increase in fruit temperature under this cover. These results suggest that this behavior could be modeled to predict the potential yield and fruit quality based on the light and temperature conditions of each type of cover, including new tools such as machine learning algorithms, which have been shown to successfully predict yield in blueberries based on computer simulation modeling datasets, with a significant impact when data are collected from different environments [18].

Since the introduction of new cultivars is necessary to meet the changing needs of consumers as well as growers, the modeling of production systems would allow reaching the highest yield and quality potential, which is particularly relevant given the findings reported in this study and the benefits of protective cover systems under the current climate change scenario. Additionally, economic analyses should be considered in further research, as cover materials such as woven covers and netting are manufactured from high-density polyethylene (HDPE), while plastics are manufactured from low-density polyethylene (LDPE), which have marked differences in manufacturing costs and useful life. In addition, it will be relevant to consider the environmental effects of these cover materials, especially HDPE, for which it will be necessary to improve how it is recycled and its transformation into others products such as pumps, valves, and pipework [60]. Frontier research should consider the development of biodegradable cover materials based on bio polymers from polysaccharides and other raw materials that have a sufficient mechanical resistance and useful life under field conditions [61]. Findings reported from the present research also will contribute to defining the specific radiometric characteristics of potential biodegradable cover materials.

## Figures and Tables

**Figure 1 plants-12-03556-f001:**
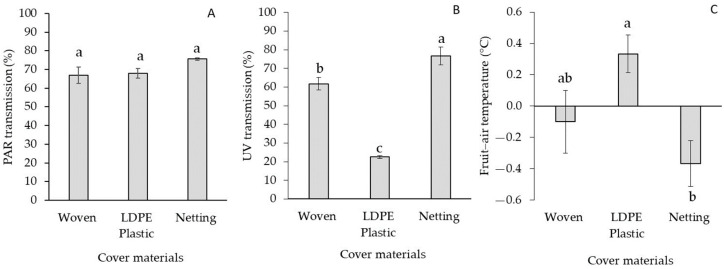
Influence of woven cover, LDPE plastic cover, and netting on PAR (**A**), UV radiation transmission (**B**), and fruit air temperature differences (**C**). Columns with different letters are statistically significant by the Tukey’s test.

**Figure 2 plants-12-03556-f002:**
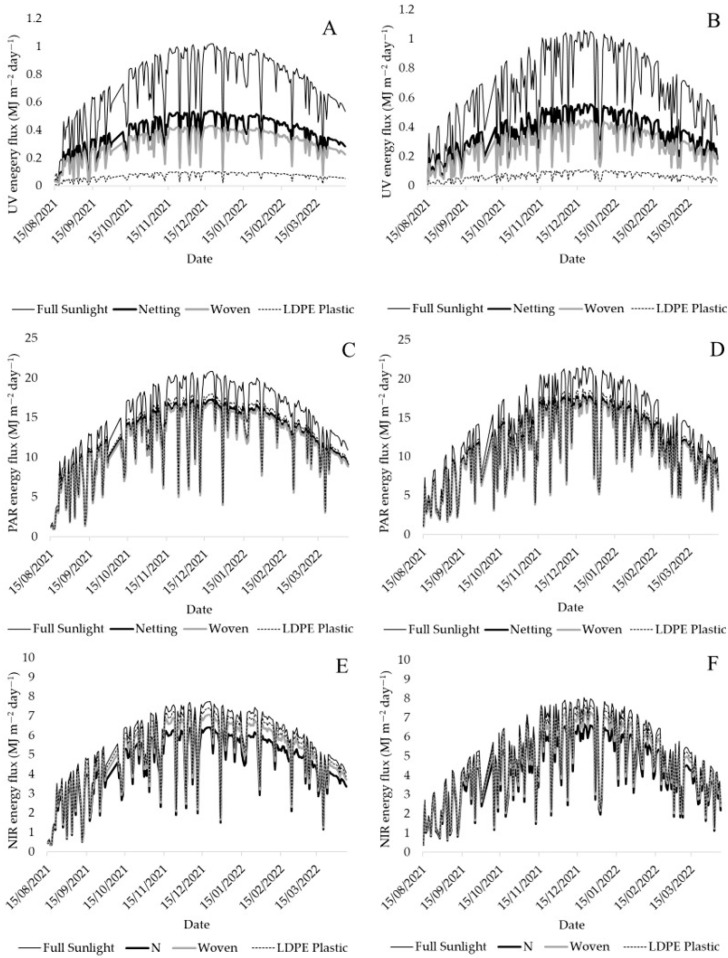
Variation in solar radiation flux in the range of UV (360–390 nm) (**A**,**B**); PAR (400–700 nm) (**C**,**D**); and NIR (700–1120 nm) (**E**,**F**) under netting, woven covers, and LDPE plastic covers in Linares and Traiguén.

**Figure 3 plants-12-03556-f003:**
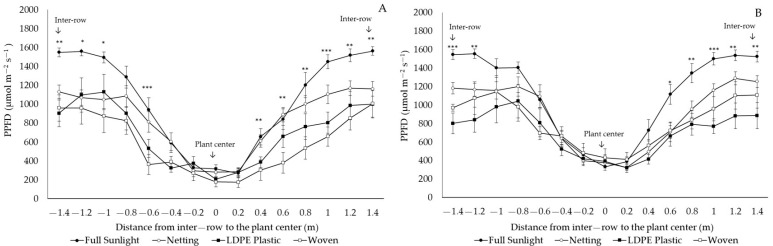
Transmission of photosynthetic photon flux density (PPFD) in ‘Legacy’ (**A**) and ‘Top Shelf’ (**B**) cultivars under netting and woven and LDPE plastic covers. Linares, Maule Region, Chile. *;**;*** significance at *p* < 0.05, 0.01 and 0.001, respectively.

**Figure 4 plants-12-03556-f004:**
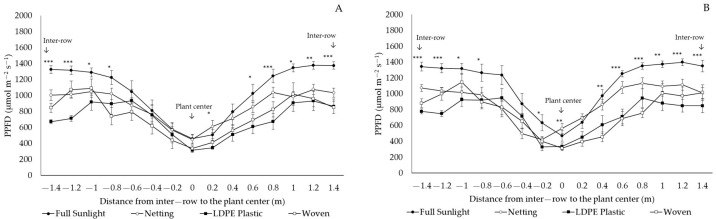
Transmission of photosynthetic photon flux density (PPFD) in ‘Legacy’ (**A**) and ‘Top Shelf’ (**B**) cultivars under netting and woven and LDPE plastic covers. Traiguén, Araucanía Region, Chile. *;**;*** significance at *p* < 0.05, 0.01 and 0.001, respectively.

**Figure 5 plants-12-03556-f005:**
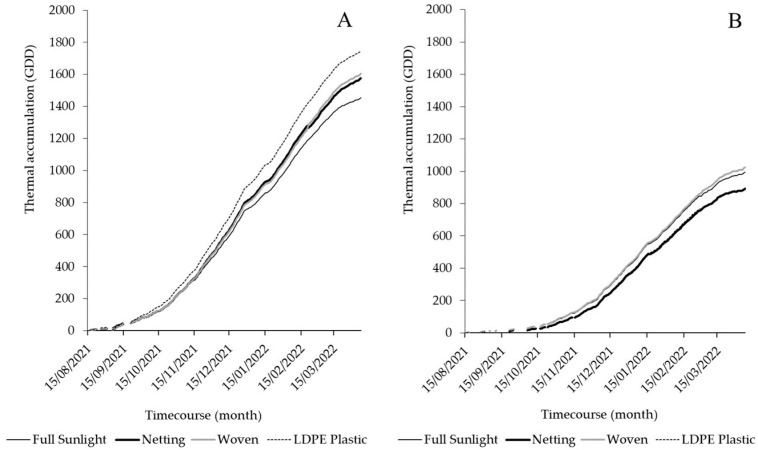
Variation in accumulated growing degree days (GDDs) under netting, and woven and LDPE plastic covers, in Linares (**A**) and Traiguén (**B**).

**Figure 6 plants-12-03556-f006:**
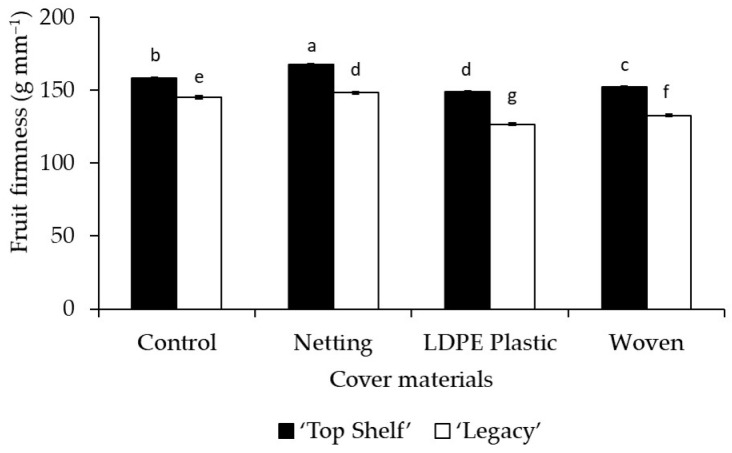
Influence of the interaction of cover materials (netting, woven, and LDPE plastic) with ‘Legacy’ and ‘Top Shelf’ cultivars on fruit firmness of blueberries in Traiguén. Columns with different letters are statistically significant by the Tukey’s test.

**Figure 7 plants-12-03556-f007:**
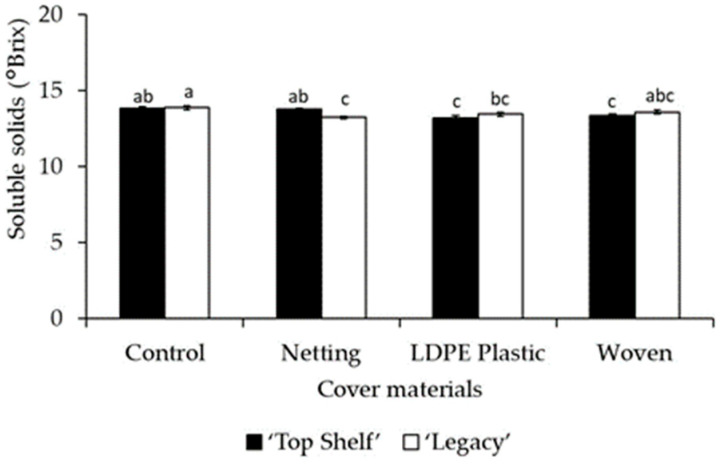
Influence of the interaction of cover materials (netting, woven, and LDPE plastic) with the ‘Legacy’ and ‘Top Shelf’ cultivars on the content of soluble solids of blueberries in Traiguén. Columns with different letters are statistically significant by the Tukey’s test.

**Figure 8 plants-12-03556-f008:**
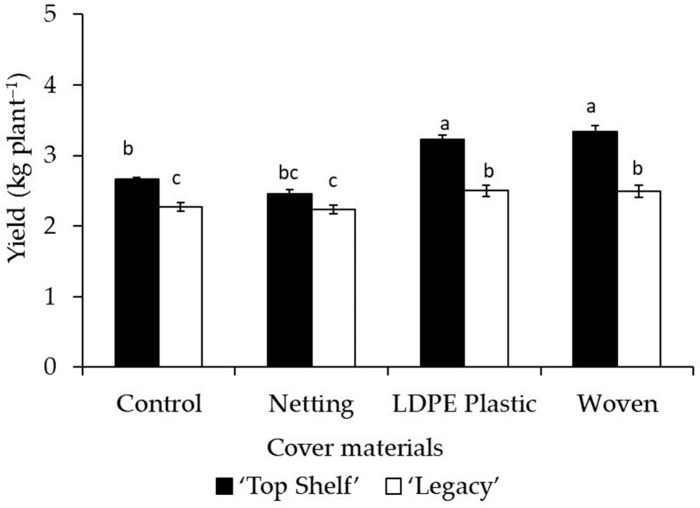
Influence of the interaction of crop cover material (netting, woven cover, and LDPE plastic cover) with ‘Legacy’ and ‘Top Shelf’ cultivars on yield in Linares. Columns with different letters are statistically significant by the Tukey’s test.

**Figure 9 plants-12-03556-f009:**
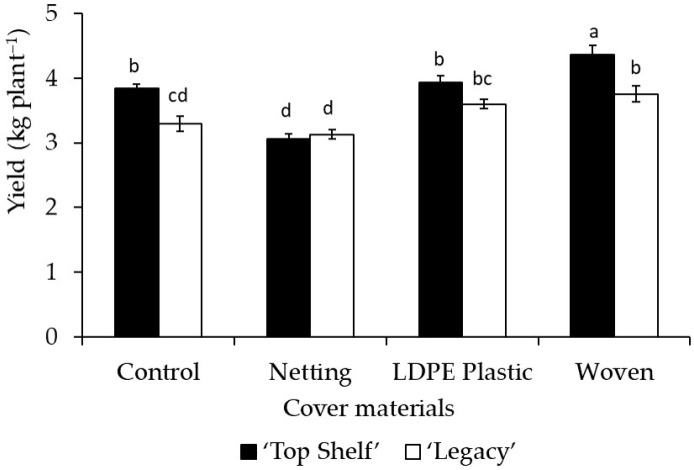
Influence of the interaction of crop cover material (netting, woven cover, and LDPE plastic cover) with ‘Legacy’ and ‘Top Shelf’ cultivars on yield in Traiguén. Columns with different letters are statistically significant by the Tukey’s test.

**Figure 10 plants-12-03556-f010:**
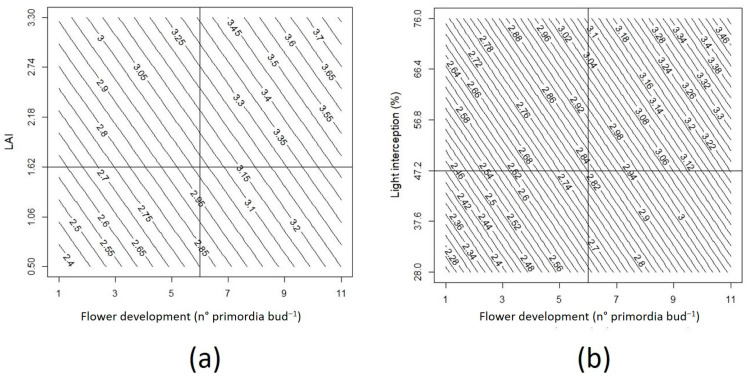
Graphic representation of the fit model for the multiple linear regression analysis between yield, flower development, and leaf area index (LAI) (regression model: Y = 2.16 + 0.09 × Fert + 0.2 × LAI; R^2^ = 0.63*; *p* < 0.05) (**a**) and between yield, flower development, and light interception (regression model: Y = 1.87 + 0.08* Fert + 0.01*Interception; R2 = 0.64*; *p* < 0.05) (**b**) as affected by covers and cultivars. Numbers on contour graph lines represent yield values in kg plant^−1^.

**Figure 11 plants-12-03556-f011:**
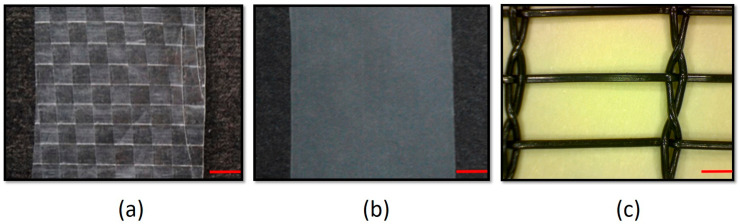
Details of woven (**a**), LDPE plastic (**b**), and net (**c**) cover materials used in protected cultivation trials in blueberries. Red bars = 1 mm.

**Figure 12 plants-12-03556-f012:**
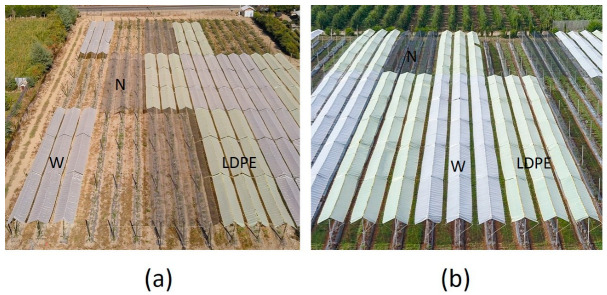
Details of installation of woven (W), LDPE plastic (LDPE), and net (N) cover materials in roof-type structure systems at Linares (**a**) and Traiguén (**b**).

**Table 1 plants-12-03556-t001:** Influence of netting, woven and LDPE plastic covers, and cultivar on light interception and leaf area index (LAI) in blueberry plants.

Treatment	Light Interception		LAI	
	(%)			
	**Linares**	**Traiguén**	**Linares**	**Traiguén**
	**Cover Materials (Cm)**			
Control	46.18 c	37.21 b	1.78 b	1.11 b
Netting	56.3 b	49.73 a	2.05 ab	1.4 a
LDPE Plastic	66.34 a	58.85 a	2.47 a	1.72 a
Woven	66.65 a	56.8 a	2.48 a	1.67 a
*p*-value	<0.0001 ***	<0.0001 ***	<0.0001 ***	<0.0001 ***
	**Cultivar (Cv)**			
Top Shelf	56.54 b	49.74 a	2.02 b	1.45 a
Legacy	61.2 a	51.55 a	2.37 a	1.5 a
*p*-value	0.0016 **	0.1429 ns	0.0029 **	0.4643 ns
*p*-value Cm × Cv	0.059 ns	0.6062 ns	0.1505 ns	0.8484 ns

Rows with different letters are statistically significant by the Tukey’s test. ns; **;*** non significance and significance at *p* < 0.01 and 0.001, respectively.

**Table 2 plants-12-03556-t002:** Influence of netting, woven and LDPE plastic covers, and cultivar on yield, fruit diameter, firmness, soluble solids (SS), acidity (A), and SS/A ratio in blueberries grown in Linares.

Treatment	Yield	Diameter	Firmness	Soluble Solids (SS) (°Brix)	Acidity (A)	SS/A Ratio
	(kg Plant^−1^)	(mm)	(g mm^−1^)		(% Citric Acid)	
	**Cover Materials (Cm)**					
Control	2.466 b	15.750 a	147.500 b	14.401 b	0.331 a	57.816 a
Netting	2.344 b	16.250 a	152.813 a	14.036 c	0.338 a	46.699 a
LDPE Plastic	2.863 a	16.750 a	138.625 d	14.604 a	0.348 a	44.893 a
Woven	2.911 a	16.125 a	143.063 c	14.394 b	0.376 a	47.666 a
*p*-value	<0.0001 ***	0.0858 ns	<0.0001 ***	<0.0001 ***	0.8907 ns	0.4418 ns
	**Cultivar (Cv)**					
Top Shelf	2.918 a	17.094 a	145.969 a	14.547 a	0.332 a	48.983 a
Legacy	2.374 b	15.344 b	145.031 a	14.171 b	0.365 a	49.554 a
*p*-value	0.0004 ***	0.0009 ***	0.1298 ns	0.0055 **	0.2775 ns	0.9446 ns
*p*-value Cm × Cv	0.0011 **	0.1434 ns	0.4779 ns	0.4613 ns	0.6198 ns	0.3320 ns

Rows with different letters are statistically significant by the Tukey’s test. ns; **;*** non significance and significance at *p* < 0.01 and 0.001, respectively.

**Table 3 plants-12-03556-t003:** Influence of netting, woven and LDPE plastic covers, and cultivar on yield, fruit diameter, firmness, soluble solids (SS), acidity (A), and SS/A ratio in blueberries grown in Traiguén.

Treatment	Yield	Diameter	Firmness	Solid Solubles (SS) (°Brix)	Acidity (A)	SS/A Ratio
	(kg Plant^−1^)	(mm)	(g mm^−1^)		(% Citric Acid)	
	**Cover Materials (Cm)**					
Control	3.566 b	15.250 a	151.750 b	13.836 a	0.434 a	33.643 a
Netting	3.098 c	14.750 a	157.813 a	13.509 b	0.489 a	31.080 a
LDPE Plastic	3.766 b	15.375 a	137.750 d	13.324 b	0.393 a	35.509 a
Woven	4.057 a	15.438 a	142.188 c	13.484 b	0.424 a	35.222 a
*p*-value	<0.0001 ***	0.3068 ns	<0.0001 ***	0.0066 **	0.4880 ns	0.6371 ns
	**Cultivar (Cv)**					
Top Shelf	3.800 a	16.688 a	156.531 a	13.549 a	0.447 a	32.439 a
Legacy	3.443 b	13.719 b	138.219 b	13.527 a	0.423 a	35.288 a
*p*-value	0.0155 *	0.0034 **	<0.0001 ***	0.6279 ns	0.5561 ns	0.4101 ns
*p*-value Cm x Cv	0.0500 *	0.7350 ns	0.0007 ***	0.0179 *	0.8059 ns	0.8764 ns

Rows with different letters are statistically significant by the Tukey’s test. ^ns^; *; **;*** non significance and significance at *p* < 0.05, 0.01 and 0.001, respectively.

**Table 4 plants-12-03556-t004:** Influence of netting, woven and LDPE plastic covers, and cultivar on flower development and fruit set in blueberries.

Treatment	Flower Development		Fruit Set	
	(n° Primordia Bud^−1^)		(%)	
	**Linares**	**Traiguén**	**Linares**	**Traiguén**
	**Cover Materials (Cm)**			
Control	6.7 a	7.3 a	75.5 a	77.3 b
Netting	6.8 a	7.8 a	77.5 a	85.7 a
Plastic	7.3 a	7.8 a	77.4 a	78.6 b
Woven	6.6 a	7.7 a	76.5 a	80 ab
*p*-value	0.1402 ns	0.4754 ns	0.9033 ns	0.0183 *
	**Cultivar (Cv)**			
Top Shelf	7.6 a	8.3 a	77.1 a	80.7 a
Legacy	6.1 b	7.1 b	76.4 a	80.1 a
*p*-value	<0.0001 ***	0.0003 ***	0.8424 ns	0.8559 ns
*p*-value Cm × Cv	0.542 ns	0.7244 ns	0.6607 ns	0.9429 ns

Rows with different letters are statistically significant by the Tukey’s test. ns; *; *** non signifi-cance and significance at *p* < 0.05 and 0.001, respectively.

## Data Availability

The data presented in this study are available on request from the corresponding author.

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
