# Peer review of "Rain Cover and Netting Materials Differentially Affect Fruit Yield and Quality Traits in Two Highbush Blueberry Cultivars via Changes in Sunlight and Temperature Conditions"

_plants, 2023, doi:10.3390/plants12203556_

Round 1

Reviewer 1 Report

Figure 4 - is this for Traiguen or Linares? The figure caption seems to indicate Linares, but the text (lines 128-130) indicate Traiguen. 

Lines 137 - 138: For LAI, woven and LDPE are not significantly different from netting as claimed in lines 137-138. 

Line 244 - "parameter" should be replaced with "variable'. 

Author Response

Dear Reviewer 1

It is pleasure to indicate that revisions required were incorporated in the manuscript plants-2634747 as follows:

Figure 4 - is this for Traiguen or Linares? The figure caption seems to indicate Linares, but the text (lines 128-130) indicate Traiguen.

Yes. It was corrected the figure caption by “Traiguen”.  Line 136 – 137.

Lines 137 - 138: For LAI, woven and LDPE are not significantly different from netting as claimed in lines 137-138.

Yes. It was corrected. Line 139.   

Line 244 - "parameter" should be replaced with "variable'.

Was replaced. Line 145.

Kind regards

Dr. Richard M. Bastías

Reviewer 2 Report

In this manuscript (plants-2634747) entitled "Rain Cover and Netting Materials differentially affect fruit yield and quality traits in two highbush blueberry cultivars by changes in sunlight and temperature conditions" submitted to Plants, María F. Matamala and colleagues have evaluated the simultaneous effect of LDPE plastic cover, woven cover and netting materials on environmental components (UV light, PAR, NIR and growing degree days, GDD), plant performance (light interception, leaf area index, LAI, yield and flower development), and fruit quality traits (firmness, total soluble solids and acidity) in two blueberry cultivars. The type of cover was demonstrated to differentially affect yield and fruit quality in blueberries due to the specific light and temperature conditions generated under these materials. This research is interesting and convincing, but minor points need to be addressed to improve the quality of this manuscript.

1. For Figure 11, pictures to show the rain cover and netting materials employed in this study have been presented, but the scale bars should be included

2. For Figure 11, the covers were installed on a roof-type structure, and the picture of this roof-type structure should be included here.

3. For Figures 3 and 4, the term ‘distance to centers’ should be explained in the legend.

4. For Figure 5, please replace the ‘date’ with a timecoure ‘month’ in the x-axis.

Author Response

Dear Reviewer 2

It is pleasure to indicate that revisions required were incorporated in the manuscript plants-2634747 as follows: 

  1. For Figure 11, pictures to show the rain cover and netting materials employed in this study have been presented, but the scale bars should be included

Scale bars were included in the Figure 11. Lines 483 – 486.

  1. For Figure 11, the covers were installed on a roof-type structure, and the picture of this roof-type structure should be included here.

Picture with roof-type structure installation system used in the trials was included (Figure 12). Lines 496 – 500.

  1. For Figures 3 and 4, the term ‘distance to centers’ should be explained in the legend.

The term ´distance centers´ was clearly explained into the legend of Figures 2 and 4.  Line 139 – 145.

  1. For Figure 5, please replace the ‘date’ with a timecoure ‘month’ in the x-axis.

Was replaced in Figure 5. Line – 175 – 176

Kind regards

Dr. Richard M. Bastías

Reviewer 3 Report

This paper investigates the effects of different rain cover and netting materials on blueberry yield and quality. The study found that the type of cover material used can impact the transmission of UV and NIR light, accumulation of growing degree days, and fruit firmness. Greater firmness was observed under netting, while significantly lower values were recorded in fruits grown under LDPE plastic cover. The research suggests that modeling production systems based on light and temperature conditions of each type of cover can help predict potential yield and fruit quality. I would suggest revisions as listed below

1.      What are the economic and environmental implications of using different types of cover materials in blueberry production?

2.      How can predictive models based on environmental variables be used to optimize blueberry production and reduce risk for growers?

3.      How do changes in light and temperature conditions impact blueberry growth and development, and how can this information be used to improve production systems?

4.      In the introduction, the authors should also mention the recent work using machine learning for environmental science, some references to build the related work

https://www.sciencedirect.com/science/article/pii/S0022169419303026

https://www.mdpi.com/2072-4292/14/13/3228

The english is fine

Author Response

Dear Reviewer 3

It is pleasure to indicate that revisions required were incorporated in the manuscript plants-2634747 as follows: 

  1. What are the economic and environmental implications of using different types of cover materials in blueberry production?

A paragraph was included (conclusions section) in the manuscript with an analysis of future research challenges from an economic and environmental perspective, and with emphasis in the manufacturing costs, useful life, recycling systems and bio-degradable polymer options for cover materials.  Lines 600 – 610.

  1. How can predictive models based on environmental variables be used to optimize blueberry production and reduce risk for growers?

A paragraph was included (discussion section) in the manuscript with a background of predictive structural-functional models to optimize crop production based in environmental changes in the light and temperature under greenhouse conditions, and with emphasis in its application to blueberry productions under covers.  Lines 384  - 390.

  1. How do changes in light and temperature conditions impact blueberry growth and development, and how can this information be used to improve production systems?

A paragraph was included (discussion section) in the manuscript with a background about of light and temperature requirements to reach optimal photosynthesis, growth and development in blueberries, and its application to develops of models to predict yield under different cover materials. Lines 377 - 384.

  1. In the introduction, the authors should also mention the recent work using machine learning for environmental science, some references to build the related work

https://www.sciencedirect.com/science/article/pii/S0022169419303026

https://www.mdpi.com/2072-4292/14/13/3228

A paragraph was included (introduction section) in the manuscript with a background about potential use of machine learning algorithms to predict yield and quality in blueberries under protected systems. Lines 87 – 94.

Also, a paragraph in the same direction was included in the conclusions. Lines 592 – 595.  

Kind regards

Dr. Richard M. Bastías
